# Circulating Protein Biomarkers for Prognostic Use in Patients with Advanced Pancreatic Ductal Adenocarcinoma Undergoing Chemotherapy

**DOI:** 10.3390/cancers14133250

**Published:** 2022-07-01

**Authors:** Sidsel C. Lindgaard, Emil Maag, Zsófia Sztupinszki, Inna M. Chen, Astrid Z. Johansen, Benny V. Jensen, Stig E. Bojesen, Dorte L. Nielsen, Zoltan Szallasi, Julia S. Johansen

**Affiliations:** 1Department of Oncology, Copenhagen University Hospital—Herlev and Gentofte, DK-2730 Herlev, Denmark; inna.chen@regionh.dk (I.M.C.); astrid.zedlitz.johansen@regionh.dk (A.Z.J.); bvittrup@dadlnet.dk (B.V.J.); dorte.nielsen.01@regionh.dk (D.L.N.); julia.sidenius.johansen@regionh.dk (J.S.J.); 2BioXpedia, DK-8200 Aarhus N, Denmark; emil@bioxpedia.com (E.M.); zoltan.szallasi@childrens.harvard.edu (Z.S.); 3Danish Cancer Society Research Center, DK-2100 Copenhagen, Denmark; zmsz@cancer.dk; 4Department of Clinical Biochemistry, Copenhagen University Hospital—Herlev and Gentofte, DK-2730 Herlev, Denmark; stig.egil.bojesen@regionh.dk; 5Department of Clinical Medicine, Faculty of Health and Medical Sciences, University of Copenhagen, DK-2200 Copenhagen, Denmark; 6Computational Health Informatics Program, Boston Children’s Hospital, Harvard Medical School, Boston, MA 02115, USA; 7Department of Medicine, Copenhagen University Hospital—Herlev and Gentofte, DK-2730 Herlev, Denmark

**Keywords:** biomarkers, inflammation, pancreatic cancer, prognosis, protein arrays

## Abstract

**Simple Summary:**

We investigated the connection between the levels of 92 circulating proteins and survival in patients with advanced pancreatic ductal adenocarcinoma. Serum samples from 363 patients with advanced pancreatic ductal adenocarcinoma were examined using the Olink Immuno-Oncology panel. Two protein signatures were found containing seven and four proteins, respectively. These two protein indices discriminated patients with very short overall survival (<90 days) from patients with long overall survival (>2 years), with AUC values of 0.97–0.99 in the discovery cohort, and 0.89–0.82 in the replication cohorts. Further analyses were conducted exploring early changes in protein levels, and protein expression in different treatment groups.

**Abstract:**

Patients with advanced pancreatic ductal adenocarcinoma (PDAC) have a dismal prognosis. We aimed to find a prognostic protein signature for overall survival (OS) in patients with advanced PDAC, and to explore whether early changes in circulating-protein levels could predict survival. We investigated 92 proteins using the Olink Immuno-Oncology panel in serum samples from 363 patients with advanced PDAC. Protein panels for several survival cut-offs were developed independently by two bioinformaticians using LASSO and Ridge regression models. Two panels of proteins discriminated patients with OS < 90 days from those with OS > 2 years. Index I (CSF-1, IL-6, PDCD1, TNFRSF12A, TRAIL, TWEAK, and CA19-9) had AUCs of 0.99 (95% CI: 0.98–1) (discovery cohort) and 0.89 (0.74–1) (replication cohort). For Index II (CXCL13, IL-6, PDCD1, and TNFRSF12A), the corresponding AUCs were 0.97 (0.93–1) and 0.82 (0.68–0.96). Four proteins (ANGPT2, IL-6, IL-10, and TNFRSF12A) were associated with survival across all treatment groups. Longitudinal samples revealed several changes, including four proteins that were also part of the prognostic signatures (CSF-1, CXCL13, IL-6, TNFRSF12A). This study identified two circulating-protein indices with the potential to identify patients with advanced PDAC with very short OS and with long OS.

## 1. Introduction

Pancreatic ductal adenocarcinoma (PDAC) is one of the deadliest cancers globally [1]. More than 80% of patients with PDAC are diagnosed with locally advanced or metastatic disease, with a 5-year survival of only 3% [1,2,3]. The treatment options for patients with advanced disease are limited to best supportive care or palliative chemotherapy [4,5]. Palliative chemotherapy is currently one of the following: a combination of 5-flourouracil (5-FU), irinotecan, oxaliplatin, and leucovorin (modified (m)FOLFIRINOX; median overall survival (OS) of 11.1 months) [6]; a combination of gemcitabine and nab-paclitaxel (median OS of 8.5 months) [7]; or monotherapy with gemcitabine (median OS of 6 months) [8].

Historically, gemcitabine monotherapy was the only approved treatment in Denmark for advanced PDAC up until 2012 when mFOLFIRINOX was approved. In 2014, the combination of gemcitabine and nab-paclitaxel was approved. Most patients with advanced PDAC remain on first-line chemotherapy for a limited period due to treatment resistance and the quick deterioration of these patients, emphasising the difficulties in reaching effective treatment results [9].

Inflammation is a hallmark of cancer, and chronic inflammation is a risk factor for PDAC development [10,11,12]. When diagnosed, PDAC tumours are characterised by an abundant desmoplastic stroma, resulting in an immunosuppressive microenvironment; this is a major obstacle to the delivery of standard chemotherapy [13,14].

Routinely assessed circulating biomarkers of the systemic inflammatory response such as neutrophil-lymphocyte ratio (NLR), lymphocyte-monocyte ratio, C-reactive protein (CRP), the modified Glasgow Prognostic Score (CRP + albumin), and the Memorial Sloan Kettering Prognostic Score (NLR + albumin) are prognostic markers in patients with advanced PDAC, but are yet to be approved for use in routine care [15,16,17].

Plasma CA19-9 is the most-studied prognostic biomarker in patients with advanced PDAC, and a marked decrease in CA19-9 levels during treatment is associated with improved survival [18]. However, CA19-9 is not specific to PDAC, can be elevated in other diseases, and between 5% and 7% of the population do not express the enzyme necessary for making CA19-9 [19,20]. Recently, the focus of protein biomarker research has shifted from single-biomarker measurements toward using multiple biomarkers in combinations. An increasingly used technology is the antibody-based proximity extension assay (PEA), e.g., the 92-protein Immuno-Oncology (I-O) panel from Olink Proteomics (Uppsala, Sweden, www.olink.com, accessed on 22 May 2022). With PEA, pre-determined sets of proteins are measured. For each protein, there is a pair of antibody-probes labelled with oligonucleotides with a slight affinity to one another, and if both antibodies bind to the protein in close proximity, the oligonucleotides can be extended by a DNA polymerase. This forms a unique sequence acting as a surrogate for the protein. This sequence can then be quantified using quantitative real-time PCR (qPCR) [21,22]. We, and others, have demonstrated that this method can identify protein panels that could be useful for early diagnosis of PDAC [23,24]. In our previously published study, we found two protein indices for the identification of patients with PDAC from patients with non-malignant pancreatic diseases and healthy individuals. Both indices had AUC-values > 0.90 [23]. In the present study, we used the I-O panel from Olink to find a prognostic protein signature for OS in patients with advanced PDAC, and to explore whether changes in circulating-protein levels during chemotherapy could predict survival.

## 2. Materials and Methods

The study was conducted according to the REMARK (Reporting Recommendations for Tumour Marker Prognostic Studies) and TRIPOD (Transparent Reporting of a Multivariable Prediction Model for Individual Prognosis or Diagnosis) guidelines [25,26].

### 2.1. Patients

This prospective biomarker study included 737 blood samples from 363 patients with either locally advanced (*n* = 94) or metastatic PDAC (*n* = 269).

The patients were included, at five oncological departments in Denmark, in the BIOPAC study (“BIOmarkers in patients with PAncreatic Cancer—can they provide new information about the disease and improve diagnosis and prognosis of the patients?”, NCT03311776, www.herlevhospital.dk/biopac/, accessed on 22 May 2022). All patients provided written informed consent. Blood samples were drawn between February 2009 and August 2018. Patients were followed up until 19 March 2021. The patients still alive on that date (*n* = 4) had been followed for between 35 and 53 months.

### 2.2. Ninety-Two Proteins Determined using the Olink Immuno-Oncology Assay

The samples were analysed for 92 proteins using the Olink Immuno-Oncology Assay. Samples were analysed across 14 plates during the period 23 October 2018 to 26 April 2019. Eight samples from the first run of plates were included on all remaining plates for bridging purposes. The analyses were based on the PEA method [21]. PEA gives abundance levels for each protein measured as NPX values (Normalised Protein eXpression) on a log2 scale. For details and full list of proteins, see Appendix A. The PEA assay was chosen over other protein assays due to several factors. Other approaches for protein detection, such as mass spectroscopy and ELISA, have some disadvantages when measuring plasma proteins of low abundance: ELISA is not scalable to measure >90 proteins at a time per sample, and mass spectroscopy favours highly abundant proteins [27]. With PEA, it is possible to measure proteins of low abundance with high sensitivity and specificity while enabling high throughput using a minimal amount of sample. This makes the assay ideal for measuring a high range of proteins in a large number of samples [27]. The analyses were performed according to the manufacturer’s instructions at BioXpedia, Aarhus, Denmark. BioXpedia was blinded to the study endpoint, and no research questions or clinical data were revealed before all samples had been analysed.

### 2.3. Statistical Analyses

The statistical analyses and model building were exploratory, since no single validated model exists for these types of analyses. The same research questions and data (Olink results and clinical data) were given to two bioinformaticians, who, independently of one another, conducted analyses and reported results. In the first approach, missing values for CA19-9 (13 samples) were imputed, and in the second approach, the samples with missing CA19-9 values were excluded. The total number of patients in the analyses was, therefore, 363 patients in the first statistical approach, and 350 patients in the second statistical approach. Both bioinformaticians divided the cohort into discovery and replication cohorts. Patients were allocated to discovery and replication cohorts: 66% and 70% were allocated to the discovery cohort, for the first and second statistical approach, respectively. LASSO and Ridge regression models, as well as LASSO-Regularised Cox Regression models, were used for the building of prognostic models. Subgroup analyses according to treatment and survival were performed using t-test and Wilcoxon rank-sum test. Longitudinal analyses were performed using a linear mixed-effects model or Cox regression with time-dependent covariates. For more details on the statistical analyses, see Appendix A. No power calculations for appropriate study size could be made because this was an exploratory study.

## 3. Results

The patient characteristics are shown in Table 1. The median age was 68 years (range 38–88), and 74.1% had stage IV disease. The baseline median CA19-9 was 998 kU/L (IQR 132–6770 kU/L). There was a slight overweight of women in the replication cohort with the second statistical approach (Index II). This cohort also had a high median CA19-9 (2180 kU/L) compared with the other groups. OS was as expected for this patient group.

### 3.1. Pre-Treatment Plasma-Protein Levels in PDAC Patients in Relation to Survival

The first step was to explore the differential expression of the 92 proteins in the Olink Immuno-oncology panel and CA19-9. The differential expression was evaluated in four survival groups (≤90 days vs. >90 days, ≤180 days vs. >180 days, <90 days vs. >1 year, and <90 days vs. >2 years). Fifty-one of the plasma proteins had statistically significant (*p* < 0.05) differences in pre-treatment plasma levels (baseline) in one or more of these survival groups. Eight proteins (macrophage colony-stimulating factor 1 (CSF-1), hepatocyte growth factor (HGF), interleukin (IL)-6, IL-8, monocyte chemotactic protein 3 (MCP-3), placenta growth factor (PGF), tumour necrosis factor receptor superfamily member 12A (TNFRSF12A), and vascular endothelial growth factor A (VEGFA)) had *p* values < 0.001 in all four of the comparisons conducted, and are highlighted in Table 2. IL-6 was found to be the protein with the most statistically significant difference across all survival groups.

### 3.2. Prognostic Protein Panels for Very Short vs. Very Long Survival (<90 Days vs. >2 Years)

After the preliminary exploration of differentially expressed proteins, the primary objective of this study was to identify a protein panel, giving an optimal combination of markers for the discrimination between patients with very short (<90 days) and very long survival (>2 years). As two bioinformaticians worked independently of one another on this research question, we herein present two protein indices reached using slightly different statistical approaches. Details regarding the differentially expressed proteins in the comparison of survival <90 days vs. >2 years are shown in Appendix A.

With the first statistical approach, a signature containing seven proteins was deemed to be the best-performing, and is presented as Index I. For details on all candidate signatures in this statistical approach, see Appendix A. The signature consisted of the following proteins: CSF-1, IL-6, PDCD1, TNFRSF12A, TRAIL, TWEAK, and CA19-9. For full protein names, see Appendix A. Index I had AUCs of 0.99 (95% CI: 0.98–1) in the discovery cohort, and 0.89 (0.74–1) in the replication cohort. For sensitivity and specificity, see Table 3. The models were tried with age as a predictor, and the two models (with and without age added) were compared using a DeLong test. No significant change was observed for the signatures (Table 3).

With the second statistical approach, one prognostic protein signature was identified (Index II) containing four proteins, three of which were also included in index I: CXCL13, IL-6, PDCD1, and TNFRSF12A. Index II gave an AUC of 0.97 (95% CI: 0.93–1), sensitivity of 1.00 (1.00–1.00), and specificity of 0.91 (0.83–1) in the discovery cohort. In the replication cohort the AUC was 0.82 (0.68–0.96), sensitivity 0.86 (0.64–1), and specificity 0.73 (0.55–0.91).

The ROC curves, plotting Index I and Index II, and the coefficients in the two indices are shown in Figure 1.

Furthermore, a risk score was developed for Index II. This score was then compared to survival status at different time points by creating time-dependent ROC curves. This gave AUC values for predicting survival at 6, 12, 18, 24, 30, and 36 months, respectively. A Kaplan–Meier plot was made whereby all patients were divided by a risk score < median and > median. Both plots can be found in Figure 2.

As an internal validation of the proteins included in the two indices, Kaplan–Meier plots of each individual marker were made for both the discovery and replication cohorts for both indices. The Kaplan–Meier plots are shown below in Figure 3. For larger versions of the plots with details, see Appendix A.

As the proteins in the two indices are from the Olink Immuno-Oncology panel, the relation of each protein to cancer and inflammation are reported in Appendix A.

The results from the prognostic models made using different OS cut-offs can be found in Appendix A.

These results collectively show two protein indices with promising prognostic capabilities for the differentiation between patients with very short or very long OS.

### 3.3. Subgroup Analyses

We also explored whether pre-treatment plasma-protein levels were associated with prognosis when the patients were divided according to palliative chemotherapy and survival (≤180 days and >180 days). Of the 363 patients, 135 patients (37.2%) survived ≤180 days, and 228 patients (62.8%) survived >180 days. In the different treatment groups, the patients were distributed as follows: 183 patients (50.4%) received gemcitabine, of which 78 (42.6%) survived ≤180 days and 105 (57.4%) survived >180 days; 82 patients (22.6%) received gemcitabine + nab-paclitaxel, of which 34 (41.5%) survived ≤180 days and 48 (58.5%) survived >180 days; and 98 (27.0%) received mFOLFIRINOX, of which 23 (23.5%) survived ≤180 days and 75 (76.5%) survived >180 days. A Kaplan–Meier plot of survival in the three treatment groups is found in Appendix A.

All statistically significant tests can be found in Appendix A. Volcano plots illustrating the fold changes in protein levels between patients with survival ≤180 days and patients with survival >180 days, and the relation to the *p* values for each comparison, are shown in Figure 4. The first statistical approach identified four proteins that were significantly different in all four comparisons: ANGPT2, IL-6, IL-10, and TNFRSF12A. These four proteins are highlighted in yellow in Figure 4. Boxplots illustrating the protein levels of the treatment groups can be found in Appendix A.

Details on proteins that were significantly different between treatment groups and survival groups can be found in Appendix A. Overlapping proteins between the two statistical approaches were HGF, IL-6, and IL-8 in the gemcitabine group; CSF-1 and MCP-3 in the gemcitabine + nab-paclitaxel group; and ANGPT2 and IL-6 in the mFOLFIRINOX group.

### 3.4. Early Changes in Circulating-Protein Levels after Start of Palliative Chemotherapy and Survival

Early changes in circulating-protein levels after the initiation of palliative chemotherapy were tested in 234 PDAC patients with longitudinal samples available. The protein levels in baseline pre-treatment samples were compared with the levels in samples collected before the second chemotherapy cycle (2–4 weeks after baseline depending on the chemotherapy regimen) and/or at time of the first CT evaluation (approximately after 3 months). In total, 139 patients had samples at all three timepoints. Several proteins were found to have early patterns of change associated with prognosis, including four proteins that were also a part of the prognostic signatures (CSF-1, CXCL13, IL-6, TNFRSF12A). For details, see Appendix A. Results from the univariate and multivariate analyses comparing the protein levels at different time points to OS are found in Table 4. Details on the other proteins with significant changes can be found in Appendix A.

These results show that the four proteins from Table 4, which are also a part of the prognostic indices, are highly associated with survival across timepoints in both the univariate and multivariate analyses.

## 4. Discussion

We explored the prognostic potential of 92 circulating proteins + CA19-9 in a cohort of patients with locally advanced or metastatic PDAC. Two prognostic protein signatures were developed to identify patients with very short OS (<90 days) versus patients with long survival (OS >2 years) using two slightly different statistical approaches. The resulting two indices consisted of seven and four proteins, respectively. Three proteins (IL-6, PDCD1, and TNFRSF12A) overlapped between the two indices. All the proteins in the two indices have previously been described in relation to PDAC. However, to our knowledge, it is a novel observation for all proteins but IL-6 to have prognostic value for patients with advanced PDAC.

Of the proteins found in the two indices, IL-6 is the most well-described when it comes to patients with PDAC. The protein plays an important role in the development of cachexia, and high levels of circulating IL-6 are associated with increased tumour burden and short OS in patients with advanced PDAC [28,29,30]. IL-6 has been proposed to be a driver of PDAC pathogenesis through the activation of the STAT3 pathway, which is involved in the regulation of cytokine expression, resistance to apoptosis, angiogenesis, and the promotion of metastasis [28]. Reassuringly, our study confirms the prognostic value of IL-6 in patients with advanced PDAC using both Kaplan–Meier plots, and univariate and multivariate analyses across time points.

The second overlapping protein between the two indices, PDCD1, also known as programmed cell-death protein 1 (PD-1), has been widely studied, and the axis of PD-1 and its ligand, PD-L1, are considered the most successful target for immune checkpoint-blockade therapy to date [31]. PD-L1 has been shown to be a prognostic marker in several cancer types [32,33,34].

The final overlapping protein, TNFRSF12A, also known as fibroblast growth factor-inducible 14 (Fn14), and its ligand, TWEAK, show low expressions in normal tissues; however, they have also been found to be highly expressed in several solid tumours and metastases, including pancreatic cancer cell lines [35,36]. TNFRSF12A/TWEAK has been found to be prognostic in several cancer types [37,38]. All in all, although the proteins mentioned above have not previously been described as prognostic in patients with PDAC, they are well-known markers in other cancer types, lending credibility to our findings.

Furthermore, four proteins (IL-6, TNFRSF12A, CSF-1 and CXCL13) included in the two indices were found to be prognostic across time points. As mentioned, IL-6 has been well described to be prognostic in PDAC, but TNFRSF12A, CSF-1, and CXCL13 should be further investigated to evaluate their potential as prognostic markers in PDAC.

Our group recently published a study focusing on diagnostic protein biomarkers in patients with stage I–IV PDAC [23]. The same 92 proteins + CA19-9 were determined in samples from 701 patients with PDAC (which included our 363 patients with advanced PDAC), 102 patients with nonmalignant pancreatic diseases, and 180 healthy blood donors, and we identified two indices for the identification of patients with PDAC. Both CSF-1 and TRAIL were found to be part of these indices. These two proteins, therefore, seem to have both diagnostic and prognostic potential in patients with PDAC, and have previously been shown to have prognostic capabilities in other cancer types [39,40,41].

In our present study, we further explored the potential of the 92 I-O proteins and CA19-9 as prognostic biomarkers in a subgroup analysis of patients divided by the three different types of standard palliative chemotherapy administered to the patients. The findings from the gemcitabine + nab-paclitaxel group will be validated in an ongoing clinical phase II study (PACTO, ClinicalTrials.gov NCT02767557), where results are expected in 2023.

A recent smaller study by Peng et al. included 52 patients with stage III or IV PDAC divided into “good-responders” (OS ≥12 months, *n* = 26) and “limited-responders” (OS <12 months, *n* = 26) [42]. The type of chemotherapy given to patients was not reported. With a two-sample *t-*test, 37 proteins were found to be significantly different between the two groups [42]. Only one protein (granzyme H) was a part of the Olink I-O panel examined in our study, and this was found in neither our prognostic indices nor our subgroup analyses.

The main strength of our study is the inclusion of a relatively large number of patients with advanced PDAC, and the Olink PEA method used. Today, >800 articles have described this method, including more than 120 studies of patients with cancer, but only four studies of patients with PDAC [23,24,43,44]. With results from 737 samples from 363 patients, we confirmed IL-6 as a prognostic marker in PDAC; moreover, we described other proteins that had previously been reported in relation to PDAC to also have prognostic capabilities. The indices described here will need further validation, but the relation of these proteins to other types of cancer strengthens the biological plausibility of the indices in patients with PDAC.

A further strength is the robustness of the findings, as two bioinformaticians independently worked on the same dataset and reached similar results.

Our study has several limitations. The exploratory nature of our study precludes the results from immediate clinical implementation. Our PDAC cohort is, furthermore, from a relatively homogeneous Scandinavian population, and extrapolation to more heterogeneous populations may be limited. However, we are not aware of any specific population characteristics that would limit the generalizability of our results. Our population consisted mainly of stage IV patients, with only 25.9% having stage III disease. Therefore, our findings could be driven primarily by the protein levels of the patients with the highest tumour burden and, therefore, the most extreme protein levels.

Furthermore, patients treated with gemcitabine monotherapy were overrepresented in our population. This can, in part, be explained by the available treatment regimens in the period of sample collection (2009–2018). Our results are limited by the preselection of proteins by Olink, and proteins found in other studies of patients with advanced PDAC could not be examined.

Finally, the clinical applications of prognostic protein indices, such as the two in the present study, relies on thorough validation. If validated, these prognostic indices could be used for making clinical decisions in collaboration with the patient. In a patient with a very poor prognosis, the clinician could use the results of the protein measurements as a starting point for a discussion with the patient regarding treatments. The optimal treatment for these patients is debatable. This must rely on a discussion between the patient and the clinician, since no clear scientific recommendations exist [45].

## 5. Conclusions

In conclusion, we identified two circulating-protein indices potentially identifying patients with advanced PDAC with very short OS versus patients with long OS. Changes in several of these proteins during chemotherapy were also associated with survival. Further validation is needed.

## Figures and Tables

**Figure 1 cancers-14-03250-f001:**
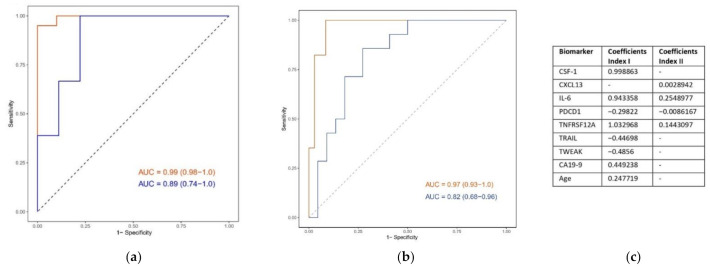
Prognostic signature, Receiver Operating Characteristics (ROC) curves. The orange lines indicate performance in the discovery cohorts, and the blue lines indicate performance in the replication cohorts: (**a**) ROC plots Index I (CSF-1, IL-6, PDCD1, TNFRSF12A); (**b**) ROC plots Index II (CXCL13, IL-6, PDCD1, TNFRSF12A); (**c**) coefficients for the biomarkers included in Index I and Index II. The three proteins overlapping between the two indices are marked in red.

**Figure 2 cancers-14-03250-f002:**
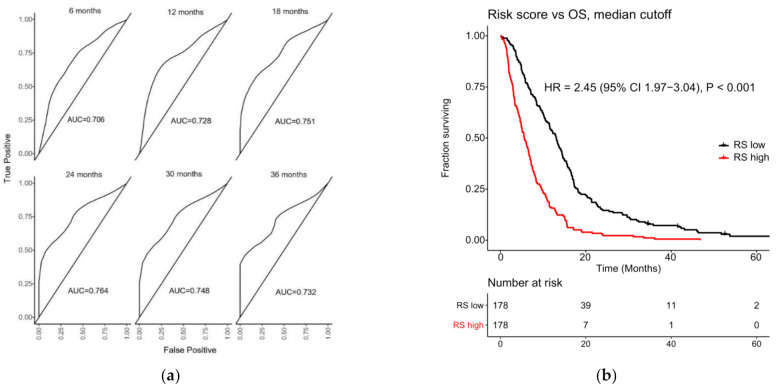
Index II: (**a**) Time-dependent ROC curves for Index II (whole cohort). Plots shown for predicting OS at 6, 12, 18, 24, 30, and 36 months, respectively. AUC values shown for each plot. Y-axes show the true positives and x-axes show the false positives; (**b**) Kaplan–Meier plot showing the risk score developed using Index II for each patient, plotted against survival. The division of patients is < or > median risk score.

**Figure 3 cancers-14-03250-f003:**
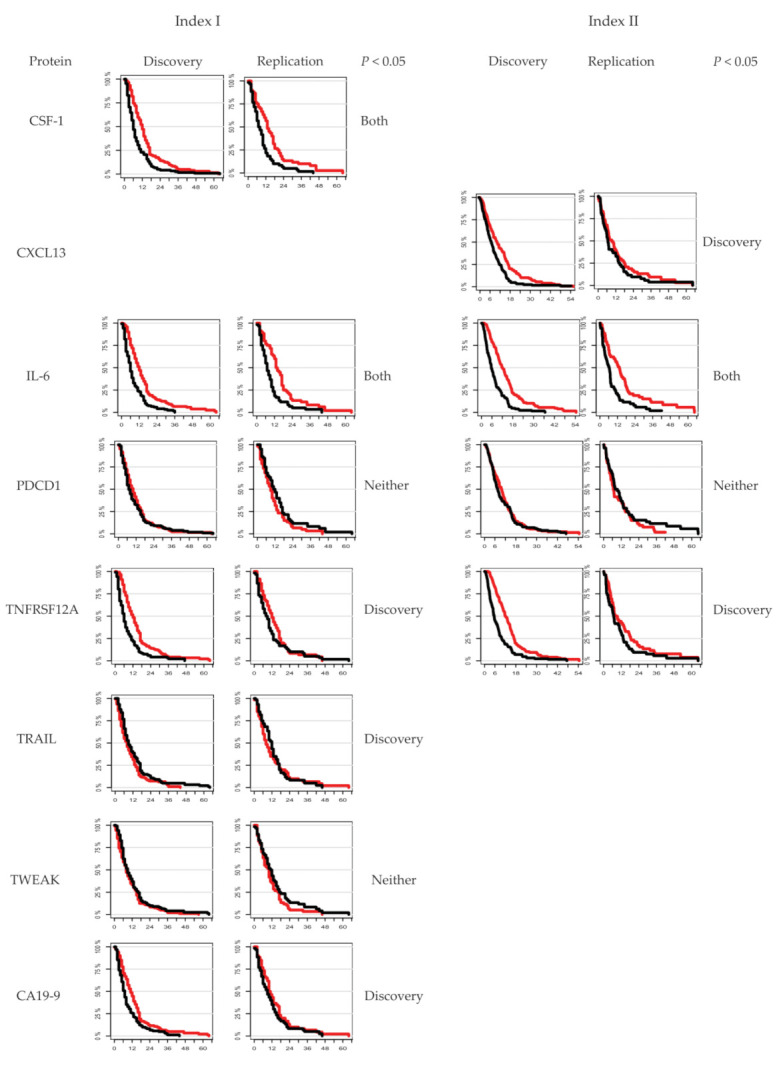
Kaplan–Meier plots showing each of the proteins included in Index I and Index II, and the individual relation to survival. Red curves illustrate <median protein levels, and black curves illustrate >median protein levels.

**Figure 4 cancers-14-03250-f004:**
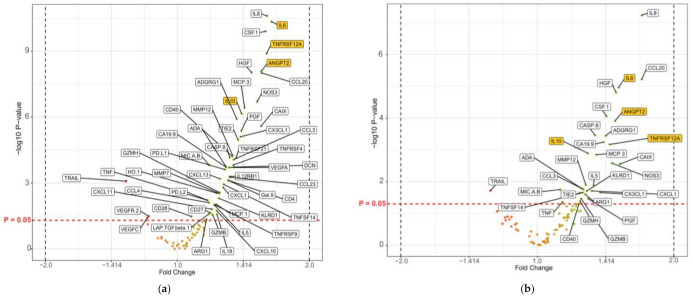
Volcano plots of the comparison of circulating-protein levels in all patients divided by survival groups (≤180 days vs. >180 days) and treatment subgroups. The relationship between non-adjusted −log10 *p* values is described on the *y*-axis and fold changes on the *x*-axis for the proteins. Proteins with significant non-adjusted *p* values (*p* < 0.05) are labelled with abbreviated names. The horizontal red dashed line represents *p* = 0.05. The x-axes represent the fold change between protein levels in the survival groups (≤180 days vs. >180 days). An example is IL-10 in Figure 4a, which has a fold change of approximately 1.4, meaning that the IL-10 NPX-values in patients surviving ≤ 180 days is approximately 1.4 times higher than in patients surviving >180 days. The x-axes have been modified from a log2 scale. (**a**) All patients divided into survival groups of ≤180 days (*n* = 135) and >180 days (*n* = 228); (**b**) patients treated with gemcitabine (survival ≤180 days (*n* = 78) vs. >180 days (*n* = 105)); (**c**) patients treated with gemcitabine + nab-paclitaxel (survival ≤180 days (*n* = 34) vs. >180 days (*n* = 48)); and (**d**) patients treated with mFOLFIRINOX (survival ≤180 days (*n* = 23) vs. >180 days (*n* = 75).

**Table 1 cancers-14-03250-t001:** Patient characteristics.

	No. (%) ^a^ of Patients				
	Index I		Index II		
	Discovery Cohort (*n* = 243)	Replication Cohort (*n* = 120)	Discovery Cohort (*n* = 257)	Replication Cohort (*n* = 106)	Total Population (*n* = 363)
Age, median (range)	68 (42–88)	68 (38–85)	68 (38–88)	68 (40–88)	68 (38–88)
≥70 years, *n* (%)	95 (39.1)	55 (45.8)	100 (38.9)	50 (47.2)	150 (41.3)
Sex, Male	133 (54.7)	65 (54.2)	147 (57.2)	51 (48.1)	198 (54.5)
Female	110 (45.3)	55 (45.8)	110 (42.8)	55 (51.9)	165 (45.5)
Stage III	69 (28.4)	25 (20.8)	64 (24.9)	30 (28.3)	94 (25.9)
IV	174 (71.6)	95 (79.2)	193 (75.1)	76 (71.7)	269 (74.1)
ECOG Performance Status 0	106 (43.6)	53 (44.2)	108 (42.0)	51 (48.1)	159 (43.8)
1	121 (49.8)	59 (49.2)	128 (49.8)	52 (49.1)	180 (49.6)
2	12 (4.9)	7 (5.8)	16 (6.2)	3 (2.8)	19 (5.2)
Unknown	4 (1.7)	1 (0.8)	5 (1.9)	0 (0.0)	5 (1.4)
Diabetes	65 (26.7)	24 (20.0)	60 (23.3)	29 (27.4)	89 (24.5)
Smoking, Former	91 (37.5)	56 (46.7)	106 (41.2)	41 (38.7)	147 (40.5)
Current	60 (24.7)	27 (22.5)	66 (25.7)	21 (19.8)	87 (24.0)
Never	80 (32.9)	34 (28.3)	72 (28.0)	42 (39.6)	114 (31.4)
Unknown	12 (4.9)	3 (2.5)	13 (5.1)	2 (1.9)	15 (4.1)
Time from diagnosis to baseline sample, days ^b^	21 (16–29)	20 (15–33)	21 (16–31)	20 (16–27)	21 (16–31)
Overall survival, months ^b^	8 (5–15)	10 (5–17)	8 (5–15)	7 (4–16)	8 (4–15)
Baseline CA19-9, kU/L ^b^	1070 (165–7285)	886 (92–5863)	840 (128–6675)	2180 (175–6770)	998 (132–6770)
Gemcitabine	126 (51.9)	57 (47.5)	127 (49.4)	56 (52.8)	183 (50.4)
Gemcitabine + nab-paclitaxel	50 (20.6)	32 (26.7)	56 (21.8)	26 (24.5)	82 (22.6)
mFOLFIRINOX	67 (27.5)	31 (25.8)	74 (28.8)	24 (22.6)	98 (27.0)

^a^ Unless otherwise noted; ^b^ median (interquartile range). Abbreviations: ECOG—Eastern Cooperative Oncology Group.

**Table 2 cancers-14-03250-t002:** Statistically significant proteins in comparisons between survival groups.

Protein	Comparison							
	≤90 Days (*n* = 57) vs. >90 Days (*n* = 306)		≤180 Days (*n* = 135) vs. >180 Days (*n* = 183)		<90 Days (*n* = 57) vs. >1 Year (*n* = 127)		<90 Days (*n* = 57) vs. >2 Years (*n* = 30)	
	*p* Value (not Adjusted)	Test	*p* Value (not Adjusted)	Test	*p* Value (not Adjusted)	Test	*p* Value (not Adjusted)	Test
ADA	–	–	4.1 × 10^−2^	Wilcoxon	9.8 × 10^−3^	*t-*test	5.4 × 10^−3^	*t-*test
ADGRG1	**4.9 × 10^−4^**	Wilcoxon	**3.7 × 10^−4^**	Wilcoxon	1.7 × 10^−3^	*t-*test	2.1 × 10^−2^	Wilcoxon
ANGPT2	3.2× 10^−3^	*t-*test	**1.4 × 10^−4^**	*t-*test	**3.1 × 10^−7^**	*t-*test	**2.2 × 10^−5^**	*t-*test
CA19-9	-	-	4.9 × 10^−2^	*t-*test	**3.8 × 10^−4^**	*t-*test	**6.0 × 10^−4^**	Wilcoxon
CAIX	1.5 × 10^−3^	Wilcoxon	**5.9 × 10^−4^**	Wilcoxon	**5.1 × 10^−4^**	*t-*test	4.6 × 10^−2^	Wilcoxon
CASP-8	-	-	4.0 × 10^−3^	Wilcoxon	1.1 × 10^−3^	*t-*test	1.4 × 10^−3^	*t-*test
CCL3	-	-	3.4 × 10^−3^	Wilcoxon	6.3 × 10^−3^	*t-*test	1.6 × 10^−3^	*t-*test
CCL20	1.1 × 10^−3^	Wilcoxon	**8.4 × 10^−5^**	Wilcoxon	**2.5 × 10^−5^**	*t-*test	2.0 × 10^−3^	Wilcoxon
CCL23	7.4 × 10^−6^	Wilcoxon	1.1 × 10^−2^	Wilcoxon	**6.4 × 10^−6^**	*t-*test	7.8 × 10^−6^	*t-*test
CD4	1.0 × 10^−2^	Wilcoxon	1.4 × 10^−2^	*t-*test	**5.8 × 10^−4^**	*t-*test	1.5 × 10^−2^	Wilcoxon
CD27	1.2 × 10^−2^	*t-*test	-	-	6.3 × 10^−3^	*t-*test	-	-
CD40	1.1 × 10^−2^	Wilcoxon	4.5 × 10^−3^	Wilcoxon	2.9 × 10^−3^	*t-*test	1.0 × 10^−2^	*t-*test
CSF-1	7.4 × 10^−6^	Wilcoxon	**3.5 × 10^−7^**	*t-*test	**2.6 × 10^−10^**	*t-*test	**1.3 × 10^−6^**	*t-*test
CX3CL1	1.3 × 10^−2^	Wilcoxon	2.0 × 10^−2^	Wilcoxon	**9.2 × 10^−5^**	*t-*test	1.7 × 10^−3^	Wilcoxon
CXCL1	3.5 × 10^−2^	Wilcoxon	1.9 × 10^−2^	*t-*test	5.7 × 10^−3^	*t-*test	2.9 × 10^−3^	Wilcoxon
CXCL11	-	-	-	-	-	-	4.6 × 10^−2^	*t-*test
CXCL13	-	-	1.7 × 10^−2^	Wilcoxon	1.5 × 10^−2^	*t-*test	2.9 × 10^−3^	*t-*test
DCN	2.1 × 10^−2^	Wilcoxon	2.4 × 10^−3^	Wilcoxon	3.2 × 10^−2^	*t-*test	-	-
Gal-9	-	-	-	-	1.1 × 10^−3^	*t-*test	5.8 × 10^−3^	*t-*test
GZMH	-	-	-	-	3.2 × 10^−2^	*t-*test	1.9 × 10^−2^	*t-*test
HGF	**9.3 × 10^−6^**	Wilcoxon	**3.6 × 10^−4^**	Wilcoxon	**4.0 × 10^−6^**	*t-*test	**1.5 × 10^−5^**	Wilcoxon
HO-1	-	-	2.0 × 10^−2^	Wilcoxon	-	-	-	-
ICOSLG	3.0 × 10^−2^	Wilcoxon	-	-	-	-	4.6 × 10^−2^	Wilcoxon
IL-6	**7.0 × 10^−9^**	Wilcoxon	**3.1 × 10^−8^**	Wilcoxon	**9.0 × 10^−12^**	*t-*test	**9.0 × 10^−12^**	*t-*test
IL-8	**2.2 × 10^−5^**	Wilcoxon	**3.4 × 10^−5^**	Wilcoxon	**5.2 × 10^−9^**	*t-*test	**1.3 × 10^−6^**	Wilcoxon
IL-10	-	-	8.8 × 10^−3^	Wilcoxon	**2.5 × 10^−4^**	*t-*test	1.2 × 10^−3^	Wilcoxon
IL-12RB1	4.6 × 10^−2^	Wilcoxon	2.7 × 10^−2^	Wilcoxon	5.4 × 10^−3^	*t-*test	2.6 × 10^−2^	*t-*test
IL18	-	-	-	-	3.2 × 10^−2^	*t-*test	-	-
KLRD	-	-	1.1 × 10^−2^	Wilcoxon	-	-	-	-
LAP TGF beta1	-	-	-	-	2.3 × 10^−2^	*t-*test	-	-
MCP-1	3.5 × 10^−3^	*t-*test	9.7 × 10^−3^	*t-*test	1.9 × 10^−3^	*t-*test	4.0 × 10^−2^	Wilcoxon
MCP-3	**2.5 × 10^−4^**	Wilcoxon	**2.7 × 10^−5^**	Wilcoxon	**6.1 × 10^−6^**	*t-*test	**1.6 × 10^−4^**	Wilcoxon
MIC-A/B	-	-	-	-	-	-	5.8 × 10^−3^	Wilcoxon
MMP12	**8.0 × 10^−5^**	Wilcoxon	3.9 × 10^−3^	Wilcoxon	**7.2 × 10^−5^**	*t-*test	2.0 × 10^−3^	*t-*test
MMP7	2.8 × 10^−3^	Wilcoxon	1.5 × 10^−3^	Wilcoxon	**2.5 × 10^−4^**	*t-*test	**3.6 × 10^−4^**	Wilcoxon
NOS3	4.6 × 10^−3^	Wilcoxon	3.0 × 10^−3^	Wilcoxon	**4.8 × 10^−5^**	*t-*test	1.5 × 10^−2^	*t-*test
PD-L1	1.2 × 10^−2^	Wilcoxon	2.1 × 10^−2^	Wilcoxon	**3.9 × 10^−4^**	*t-*test	**1.6 × 10^−4^**	*t-*test
PD-L2	-	-	2.6 × 10^−2^	Wilcoxon	-	-	-	-
PDGF subunit-B	-	-	4.5 × 10^−2^	Wilcoxon	-	-	-	-
PGF	**1.8 × 10^−4^**	Wilcoxon	**2.0 × 10^−4^**	Wilcoxon	**9.1 × 10^−5^**	*t-*test	**2.5 × 10^−4^**	Wilcoxon
TIE2	-	-	1.2 × 10^−2^	Wilcoxon	**3.6 × 10^−4^**	*t-*test	1.4 × 10^−3^	Wilcoxon
TNFRSF4	-	-	1.7 × 10^−2^	*t-*test	1.7 × 10^−3^	*t-*test	1.4 × 10^−3^	*t-*test
TNFRSF9	-	-	-	-	4.6 × 10^−2^	*t-*test	-	-
TNFRSF12A	**5.3 × 10^−6^**	Wilcoxon	**1.2 × 10^−4^**	Wilcoxon	**8.0 × 10^−9^**	*t-*test	**3.1 × 10^−7^**	Wilcoxon
TNFRSF21	3.4 × 10^−3^	*t-*test	2.2 × 10^−2^	*t-*test	1.9 × 10^−6^	*t-*test	2.1 × 10^−4^	*t-*test
TNFSF14	2.2 × 10^−2^	*t-*test	3.2 × 10^−2^	*t-*test	5.7 × 10^−3^	*t-*test	1.5 × 10^−2^	*t-*test
TRAIL	**6.7 × 10^−4^**	Wilcoxon	3.8 × 10^−2^	Wilcoxon	**2.1 × 10^−4^**	*t-*test	1.5 × 10^−3^	*t-*test
TWEAK	-	-	-	-	1.0 × 10^−2^	*t-*test	-	-
VEGFA	**7.7 × 10^−4^**	*t-*test	**7.1 × 10^−4^**	*t-*test	**3.5 × 10^−5^**	*t-*test	**6.2 × 10^−4^**	*t-*test
VEGFC	-	-	-	-	-	-	1.5 × 10^−2^	*t-*test
VEGFR-2	-	-	-	-	2.0 × 10^−3^	*t-*test	-	-

Proteins with statistically significant (*p* < 0.05) differences in plasma levels in baseline samples from all patients with advanced PDAC according to survival. Cells with **bold text** represent *p* values of <0.001. Yellow-highlighted cells indicate the eight proteins with *p* < 0.001 for all four comparisons. Abbreviations: Wilcoxon—Wilcoxon rank-sum test.

**Table 3 cancers-14-03250-t003:** Performance of the candidate prognostic protein signatures for Index I.

Signature	Discovery Cohort (*n* = 243)					Replication Cohort (*n* = 120)					Replication Cohort When Adding Age to the Model (*n* = 120)					
	AUC	BP Sens	BP Spec	PPV	NPV	AUC	BP Sens	BP Spec	PPV	NPV	AUC	BP Sens	BP Spec	PPV	NPV	DeLong Test *p* Value
Values with 95% confidence intervals in parentheses																
**1**	0.90( 0.75–1)	0.90 (0.75–1)	0.90 (0.60–1)	0.94 (0.83–1)	0.81 (0.66–1)	0.72 (0.53–0.92)	0.61 (0.33–0.88)	0.88 (0.66–1)	0.91 (0.81–1)	0.53 (0.42–0.81)	0.72 (0.53–0.92)	0.61 (0.33–0.88)	0.88 (0.66–1)	0.91 (0.81–1)	0.53 (0.42–0.81)	1
**2**	0.93 (0.80–1)	1 (1–1)	0.90 (0.70–1)	0.95 (0.86–1)	1 (1–1)	0.75 (0.56–0.94)	0.66 (0.38–0.94)	0.88 (0.55–1)	0.92 (0.78–1)	0.57 (0.42–0.87)	0.75 (0.56–0.94)	0.66 (0.38–0.94)	0.88 (0.55–1)	0.92 (0.80–1)	0.57 (0.43–0.88)	1
**3**	0.90 (0.76–1)	1 (0.75–1)	0.80 (0.60–1)	0.90 (0.83–1)	1 (0.66–1)	0.77 (0.57–0.96)	0.77 (0.50–0.94)	0.77 (0.55–1)	0.87 (0.80–1)	0.63 (0.46–0.90)	0.77 (0.59–0.96)	0.77 (0.44–0.94)	0.77 (0.66–1)	0.87 (0.80–1)	0.63 (0.46–0.88)	0.692
**4**	0.94 (0.83–1)	1 (1–1)	0.90 (0.70–1)	0.95 (0.86–1)	1 (1–1)	0.80 (0.62–0.98)	0.72 (0.50–1)	0.88 (0.55–1)	0.92 (0.80–1)	0.61 (0.46–1)	0.80 (0.62–0.98)	0.83 (0.55–1)	0.77 (0.55–1)	0.88 (0.80–1)	0.70 (0.5–1)	1
**5**	0.95 (0.84–1)	1 (1–1)	0.90 (0.70–1)	0.95 (0.86–1)	1 (1–1)	0.81 (0.64–0.98)	0.77 (0.49–1)	0.77 (0.55–1)	0.87 (0.78–1)	0.63 (0.47–1)	0.80 (0.62–0.98)	0.83 (0.50–1)	0.77 (0.55–1)	0.88 (0.80–1)	0.70 (0.47–1)	0.624
**6**	0.95 (0.84–1)	1 (1–1)	0.90 (0.70–1)	0.95 (0.86–1)	1 (1–1)	0.81 (0.64–0.98)	0.77 (0.38–1)	0.77 (0.44–1)	0.87 (0.78–1)	0.63 (0.45–1)	0.80 (0.63–0.98)	0.83 (0.44–1)	0.77 (0.55–1)	0.88 (0.78–1)	0.70 (0.46–1)	0.829
**7**	**0.99** **(0.98–1)**	**0.95** **(0.90–1)**	**1** **(0.90–1)**	**1** **(0.95–1)**	**0.90** **(0.83–1)**	**0.89** **(0.74–1)**	**1 ** **(0.72–1)**	**0.77 ** **(0.55–1)**	**0.90 ** **(0.81–1)**	**1 ** **(0.61–1)**	**0.88** **(0.73–1)**	**0.88** **(0.61–1)**	**0.77 ** **(0.55–1)**	**0.88 **c **(0.80–1)**	**0.77** **(0.53–1)**	**0.570**
**8**	0.95 (0.88–1)	1 (0.70–1)	0.80 (0.70–1)	0.90 (0.86–1)	1 (0.62–1)	0.80 (0.64–0.97)	0.66 (0.44–1)	0.88 (0.55–1)	0.92 (0.80–1)	0.57 (0.46–1)	0.77 (0.59–0.95)	0.72 (0.44–0.94)	0.88 (0.66–1)	0.92 (0.84–1)	0.61 (0.47–0.88)	0.533
**9**	0.98 (0.95–1)	1 (0.80–1)	0.90 (0.80–1)	0.95 (0.90–1)	1 (0.71–1)	0.77 (0.60–0.95)	0.61 (0.38–1)	0.88 (0.55–1)	0.91 (0.81–1)	0.53 (0.45–1)	0.77 (0.6–0.95)	0.55 (0.38–0.94)	1 (0.66–1)	1 (0.83–1)	0.52 (0.45–0.87)	1
**10**	0.96 (0.89–1)	0.95 (0.75–1)	0.90 (0.80–1)	0.95 (0.90–1)	0.90 (0.66–1)	0.80 (0.64–0.97)	0.72 (0.44–1)	0.88 (0.66–1)	0.92 (0.84–1)	0.61 (0.47–1)	0.80 (0.64–0.97)	0.72 (0.49–0.94)	0.88 (0.66–1)	0.92 (0.84–1)	0.61 (0.47–0.9)	1
**11**	0.93 (0.83–1)	0.95 (0.65–1)	0.80 (0.80–1)	0.90 (0.90–1)	0.88 (0.58–1)	0.83 (0.67–1)	1 (0.44–1)	0.55 (0.44–1)	0.81 (0.78–1)	1 (0.47–1)	0.80 (0.62–0.98)	0.77 (0.38–1)	0.77 (0.55–1)	0.87 (0.78–1)	0.63 (0.45–1)	0.347

**Table 4 cancers-14-03250-t004:** Comparison of protein levels at different timepoints vs. OS in univariate and multivariate analyses.

Protein	Baseline Sample		Before Second Treatment		Before First CT Scan (3 Months)	
	Univariate Analysis, HR, *p* Value	Multivariate Analysis, HR, *p* Value	Univariate Analysis, HR, *p* Value	Multivariate Analysis, HR, *p* Value	Univariate Analysis, HR, *p* Value	Multivariate Analysis, HR, *p* Value
CSF-1	1.85, *p* < 0.0001	1.79, *p* < 0.0001	1.33, *p* = 0.0464	1.35, *p* = 0.0502	1.57, *p* = 0.004	1.40, *p* = 0.043
CXCL13	1.44, *p* = 0.0007	1.30, *p* = 0.0234	1.42, *p* = 0.0148	1.33, *p* = 0.0663	1.66, *p* = 0.001	1.49, *p* = 0.02
IL-6	2.16, *p* < 0.0001	2.08, *p* < 0.0001	1.63, *p* = 0.0007	1.57, *p* = 0.0030	1.82, *p* = 0.0001	1.62, *p* = 0.0045
TNFRSF12A	1.67, *p* < 0.0001	1.57, *p* < 0.0001	1.72, *p* = 0.0002	1.68, *p* = 0.0008	1.85, *p* < 0.0001	1.79, *p* = 0.0004

In the multivariate analyses, the protein levels are combined with age, stage, baseline performance status, baseline CA19-9, and type of palliative chemotherapy.

## Data Availability

Data are available upon reasonable request. Data cannot be uploaded online due to regulations by the Danish Data Protection Agency.

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
