# Peer review of "Circulating Protein Biomarkers for Prognostic Use in Patients with Advanced Pancreatic Ductal Adenocarcinoma Undergoing Chemotherapy"

_cancers, 2022, doi:10.3390/cancers14133250_

Round 1

Reviewer 1 Report

The study aims to find early plasma protein signatures that predict survival in advanced pancreatic ductal adenocarcinoma. For this purpose, they use the Olink immuno-oncology panel, and determine the levels of 92 proteins in serum samples from 363 patients with advanced PDAC. Employing two subtly different statistical approaches, they have developed two potentially prognostic protein signatures that distinguish between patients with a very short OS (<90 days) versus patients with a long survival (OS >2 years).

Suggestions:

- In the statistical analysis section, paragraph “In the first approach [lines 113 to 116] varied slightly” is not clear. It is suggested that it be expressed in another way.

- Why are the n differents in the discovery and replication cohorts of index I and II?

- Correct expression “statistically significantly” throughout the manuscript

- In the results obtained in section 3.2 (“Prognostic protein panels for very short vs. very long survival (<90 days vs. >2 years”), a final sentence is necessary as a summary of the finding obtained after the multiple analyzes performed.

Author Response

Thank you for your comments and suggestions. Please see the attachment.

Reviewer 2 Report

Sidsel C. Lindgaard et al.'s article reports on the identification of different circulating proteic signatures which bear prognostic significance for a prospecitve cohort of locally advanced/metastatic PDAC patients both at baseline and on treatment. I would like to compliment the authors for the effort to reach such a significant sample size and for having addressed the still unanswerd question of outcome prediction in PDAC patients.

However, some points of discussion should be raised:

Minor points

1. In Introduction, Auhtors should refer more extensively to the PEA technology used since it is not broadly known, as they should as well briefly summarize its application to early PDAC identification's results they refer to.

2. Throughout all the article, the text's content is too fragmented between main text and supplemental material. I recognise that the authors were forced because of space limits, however I think that they should more wisely distribute the content to convey more clearly the key messages of the work.

3. English language is sometimes too informal so I would recomend to slightly revise this formal espect.

Major points

1. It is very recommended in my opinion that the authors explain why they choose to utilize the Olink test, more extensively mentioning its pros and cons compared to other circulating protein assays available.

2. I think that a comment on the putative and perspective clinical applications, thus utility, of this kind of approach if further validated is warranted. In fact, would upfront knowing a poorer prognosis lead the clinicians choose a more intensive regimen such as mFolfirinox? This aspect warrants some deepening in Discussion, since it might be a matter of debate. 

Author Response

(The authors gave the same response as above.)

Reviewer 3 Report

Please see my comments in the attached pdf file.

Author Response

Thank you for your immense work in reveiweing this manuscript. Please see the attachment for our response.

Round 2

Reviewer 3 Report

Please see attached pdf with my comments.

Author Response

Thank you very much for the feedback and comments. Please see attachment for replies.
